# Spatio-Temporal Neural Dynamics of Observing Non-Tool Manipulable Objects and Interactions

**DOI:** 10.3390/s22207771

**Published:** 2022-10-13

**Authors:** Zhaoxuan Li, Keiji Iramina

**Affiliations:** 1Graduate School of Systems Life Sciences, Kyushu University, Fukuoka 8190395, Japan; 2Faculty of Information Science and Electrical Engineering, Kyushu University, Fukuoka 8190395, Japan

**Keywords:** EEG, functional connectivity, manipulability, object observation, phase locking value

## Abstract

Previous studies have reported that a series of sensory–motor-related cortical areas are affected when a healthy human is presented with images of tools. This phenomenon has been explained as familiar tools launching a memory-retrieval process to provide a basis for using the tools. Consequently, we postulated that this theory may also be applicable if images of tools were replaced with images of daily objects if they are graspable (i.e., manipulable). Therefore, we designed and ran experiments with human volunteers (participants) who were visually presented with images of three different daily objects and recorded their electroencephalography (EEG) synchronously. Additionally, images of these objects being grasped by human hands were presented to the participants. Dynamic functional connectivity between the visual cortex and all the other areas of the brain was estimated to find which of them were influenced by visual stimuli. Next, we compared our results with those of previous studies that investigated brain response when participants looked at tools and concluded that manipulable objects caused similar cerebral activity to tools. We also looked into mu rhythm and found that looking at a manipulable object did not elicit a similar activity to seeing the same object being grasped.

## 1. Introduction

Tools play a special role among the objects that people usually come in contact with in daily life. Neuroscientists have found confirmatory evidence that using tools can lead to a lasting, discernible change on the perception of someone’s own body [1]. Furthermore, looking at a tool can also initiate a series of changes in cerebral activity. Many previous studies demonstrated that observing tools resulted in a left hemisphere advantage, while this did not occur with other objects [2,3,4]. The most popular explanation for the neural mechanism behind this phenomenon is that tools have the property of “manipulability” and their appearance suggests an associated action or movement [5,6]. In other words, it is reasonable to consider that the tool-associated cerebral activity is at least partly caused by the manipulability of the presented tools. However, in past decades, most studies compared tools with other objects—either manipulable or not (such as a chair or plane that could not be grasped by hand). Therefore, we suspected that some daily objects that can usually be grasped with human hands may also help with launching a similar cognitive process because they possess an almost similar manipulability to those of tools.

The first purpose of this study is to verify the aforementioned hypothesis. Moreover, we aimed to explore the relationship between seeing an object alone vs. seeing an object grasped by a hand. Previous studies have reported that seeing others’ hand actions causes a similar cerebral activity to executing the same action [7,8,9]. In another study, it was found that observing tools and watching others use tools share similar cerebral activities [10]. Because we assumed that objects with manipulability would lead to similar neural circuits to that of tools, it is necessary to investigate whether seeing objects alone and seeing other people grasping these objects have similar electrophysiological features.

In this study, we designed a simple experiment with visual presentation tasks and collected electroencephalography (EEG) data from volunteer participants. By analyzing the functional connectivity and time-frequency features, the similarities and differences between seeing objects alone and watching others interacting with these objects is demonstrated. Furthermore, we also discuss possible explanations for any unexpected results.

## 2. Materials and Methods

### 2.1. Experiments

**Materials**: Our hypothesis requires that the objects used as the stimuli need to be manipulable but are not tools. Additionally, a previous visual–somatosensory cross-modal study reported that objects from different categories may not lead to the same neural activity. Therefore, we chose only three objects that often appear in daily life, are easy to hold by hand, and do not have immediate associations with each other. Meanwhile, this design allowed us to use the same stimuli a number of times before participants felt tired. When creating the condition of “seeing an object being grasped” (i.e., participants saw an interaction with an object), to control the variables as much as possible, the conception of an interaction was analyzed first. An interaction includes three elements: subject, object, and a solution to draw a relation between them. Therefore, two more kinds of stimuli were added between “object” and “interaction”: in our design, we used a normal human hand as a subject; orange, bottle, and smart phone as objects; and hand grasping as the solution, which is one of the most common forms of manipulability in our daily life. Figure 1a shows the images used as visual stimuli in the experiment.

**Participants**: A total of 20 healthy humans (including 8 females; mean age 24.05 years, range 22–27 years) with normal or corrected-to-normal vision participated in this experiment. This study was reviewed and approved by the Department of Informatics, Faculty of Information Science and Electrical Engineering, Kyushu University (admission No. 2021-13), and every participant signed the informed consent form voluntarily before the experiment began. As all volunteers were right-handed, in this paper, we do not discuss the situation containing the left hand as a stimulus.

**Stimulus presentation**: Visual stimuli were presented to participants on a 17-inch LCD display. The resolution and refresh rate were set at 1280 × 720 pixels and 60 Hz, respectively. The distance between the eyes and display was in the range of 90–100 cm. Two runs were executed for each participant, and each run included three sessions with different topics: orange session, bottle session, and smart phone session. At the beginning of each run, the sequence of the three sessions was decided randomly. In 140 trials for each session, images containing the chosen object (five images from conditions A, C, and D) and subject (two images from condition B) were shown randomly and repetitively (20 times each image) after a fixed cross sign at the center of the screen and then back to a black screen, shown after 1 s, as depicted in Figure 1b. An interval with a duration of 1000–2000 ms was randomly placed between two trials.

### 2.2. Data Analysis

**EEG data processing**: Data from nineteen participants were included for analyses; data for one were excluded due to an unexpected technical malfunction. The recorded data were re-referenced to a common average, and then sent through a zero-phase-shift frequency domain bandpass filter with the cut-off frequency set at 1 and 30 Hz. Next, the Independent Component Analysis (ICA) completed by the Algorithm for Multiple Unknown Signals Extraction [11] was used to remove EOG artifacts. Trials with potentials over 100 µV were seen as abnormal and abandoned. Finally, over 97.5% of trials of each condition remained for further analysis. The data recorded from 200 ms before stimulus onset (as the baseline) to the end of a trial were extracted as an epoch.

**Statistical test based on Monte Carlo method**: Most of the statistical analysis revealed that the data were not normally distributed; therefore, we chose one-tailed nonparametric test methods for this research. Many researches have proven that the permutation test is reliable for testing neural signals [12,13]. In this research, the workflow can be described as follows:1.For two independent sample sets, sampA
and sampB, where $H_{0}:\;\bar{sampA}\;\leq \;\bar{sampB}$, $v_{0}$ was calculated as follows:(1)$$v_{0}=\bar{sampA}-\bar{sampB},$$
where $H_{0}$ is the null hypothesis and $v_{0}$ is the test statistic.

2.sampA and sampB were put into the same group. Then, the elements of this group were randomly divided into two sub-groups: $sampA_{1}$ and $sampB_{1}$, which had the same size. The new statistic of test $v_{1}$ was calculated as follows:


(2)
$$v_{1}=\bar{sampA_{1}}-\bar{sampB_{1}},$$


3.Step b was repeated 10,000 times to obtain $v_{1},\;v_{2},\;\ldots ,\;v_{10,000}$;

4.The $v_{1},\;v_{2},\;\ldots ,\;v_{10,000}$ values were sorted in ascending manner, and the sequence number of the first value that was greater than $v_{0}$ was identified as the “location ”.The *p*-value of the statistic test was calculated as follows:


(3)
$$p=1-\frac{location}{10,000},$$


Similarly, when it comes to a paired test, we used the bootstrap resampling approach to obtain the confidence interval of the difference between the paired samples. The bootstrap statistical method is also a nonparametric approach with proven validity and has been approved in many studies [14,15,16]. The procedures are shown below:1.For two paired sample sets, sampC and sampD, where $H_{0}:\;\bar{sampC}\;\leq \;\bar{sampD}$, we constructed a paired sample set sampP, as follows:
(4)sampP=sampC−sampD,

2.Resampling was performed from sampP with a replacement to generate a new sample set, $sampP_{1}$; then, its mean value $A_{1}$ was calculated as follows:


(5)
$$A_{1}=\bar{sampP_{1}},$$


3.The last step was repeated to obtain $A_{2},\;A_{3},\ldots ,\;A_{10,000}$, which were then sorted in ascending manner, and then, the index of the first value that was greater than zero was identified as the index. The *p*-value of this test was calculated as follows:


(6)
$$p^{\prime }=\frac{index}{10,000},$$


**Functional connectivity and effective phase-locking value (ePLV)**: We estimated the phase-locking values (PLVs) to measure the connectivity between the data recorded near the occipital lobe (a fusion of EEG recorded from electrodes Oz, O1, O2, POz, PO3, and PO4) and all the other electrodes [17]. The result of the Hilbert Transform (HT) of each epoch was used to generate analytic signals for computing the instantaneous phase at each moment. The PLV between regions i and j at time t is estimated as follows:(7)$$PLV_{i,j,t}=\sqrt{\begin{array}{c} {\left[\frac{1}{n}\overset{n}{\underset{k=1}{\sum }}\cos\left(\theta_{i,k,t}-\theta_{j,k,t}\right)\right]}^{2}\\ +{\left[\frac{1}{n}\overset{n}{\underset{k=1}{\sum }}\sin\left(\theta_{i,k,t}-\theta_{j,k,t}\right)\right]}^{2} \end{array}},$$
where n is the number of epochs and θ is the phase in radians obtained from HT [18]. For each subject, one PLV time series was estimated. However, these values do not always mean that there is a relationship between the two regions because even noise signals would have a PLV between 0 and 1. To know which of them are significantly different from the baseline (effective PLV, ePLV), estimated PLVs were submitted to a bootstrap-resampling-based, paired statistical test program to eliminate false positives by testing with the PLVs during baseline. This program works in two steps: (i) for each participant, the PLVs during the baseline period (i.e., before the stimulus was given) were resampled to extract the mean value according to central-limit theorem, and then (ii) paired tests between PLVs at each moment and the mean value were conducted. The workflow can be described as follows:

For each PLV time series,

Values during the baseline period were extracted and were put into the baseline vector;Resampling was performed from baseline with a replacement to obtain a new vector $baseline^{’}$ with the same size;The mean $baseline^{’}$ across time was calculated;Steps 2–3 were repeated 10,000 times and then a grand mean value of the results in step 3 was obtained.

After the above procedures were executed on every PLV time series, a mean value vector was generated as the baseline, which was used as a sample set of the control group in the following paired test. Finally, we could determine which of the PLVs represented a meaningful functional connectivity and could be considered as ePLVs.

**Event-related spectral perturbation (ERSP)**: Every epoch was conducted with continuous Morlet wavelet transform to unfold their frequency dimension via the Wavelet Toolbox in MATLAB (MathWorks, Natick, MA, USA). ERSP reflects the energy changes in EEG after providing a stimulus, which is defined as the ratio of power at the current time and baseline mean [19]. For each epoch, ERSP at time i of a specific frequency component j can be calculated as follows:(8)$$ERSP_{i,j}=10\times \log_{10}\frac{u_{i,j}}{baseline_{j}},$$
where $u_{i,j}$ is the absolute value of potential at time i and frequency j, and $baseline_{j}$ is the average of the one at frequency j before the stimulus was presented. To highlight the source of ERSP variation at the sensor level, a finite difference-based spatial Laplacian transformation was conducted via Brainstorm [20,21,22]. This procedure used the ERSP data to replace the potential data in the algorithm [23].

## 3. Results

### 3.1. Functional Connectivity

It is noticed that functional connectivity estimated by EEG filtered at different bands is totally inconsistent [24]. Therefore, the preprocessed EEG epochs were filtered into four different bands (delta: 1–3 Hz, theta: 4–7 Hz, alpha: 8–13 Hz, and beta: 14–30 Hz); next, the PLVs between EEG recorded at the occipital lobe and other locations were then calculated and the ePLVs were then screened out. The number of ePLVs varied over time. The topography shown in Figure 2 displays the distribution of ePLVs calculated with data from the four frequency bands at different moments. These moments were selected to show as many connections as possible. To highlight the common and different regions that were connected to the occipital lobe, the connections observed when participants saw images of interactions were overlaid on top of those for participants presented with images of objects.

At the delta and alpha bands, the number of ePLVs was fewer than that of the other bands; furthermore, across the three objects, there was a noteworthy change in the moment that the maximum number of connections appeared. By contrast, the ePLVs estimated at the theta band and the beta band were more credible because of the number of observed connections, especially their stability across time and objects. 

There were much more functional connections observed when the images from condition D were presented to participants. The results at the theta band suggested a common region including the right frontal lobe (RF), the bilateral central sulcus (L/RCS), and the right angular gyrus (RAG) whenever participants saw objects or interactions. The connection between the occipital lobe and the area covered by electrodes F5, F7, FC5, and FT7 seemed much clearer when seeing interactions than when seeing objects, and so did the left angular gyrus (LAG). Additionally, the moment that a maximum connection number appeared showed a regular pattern: “seeing objects being grasped by human hand” established more connections at earlier. The above results also supported the opinion that the theta band has advantages in observing functional connectivity [25,26,27].

Beta band ePLVs commonly appeared at both the central frontal lobe (CF) and RAG (near electrode P4 or P6) at the end of a trial, robustly. Meanwhile, the difference between seeing objects and seeing interactions is uncertain; their exclusive regions varied across objects. 

**Figure 2 sensors-22-07771-f002:**
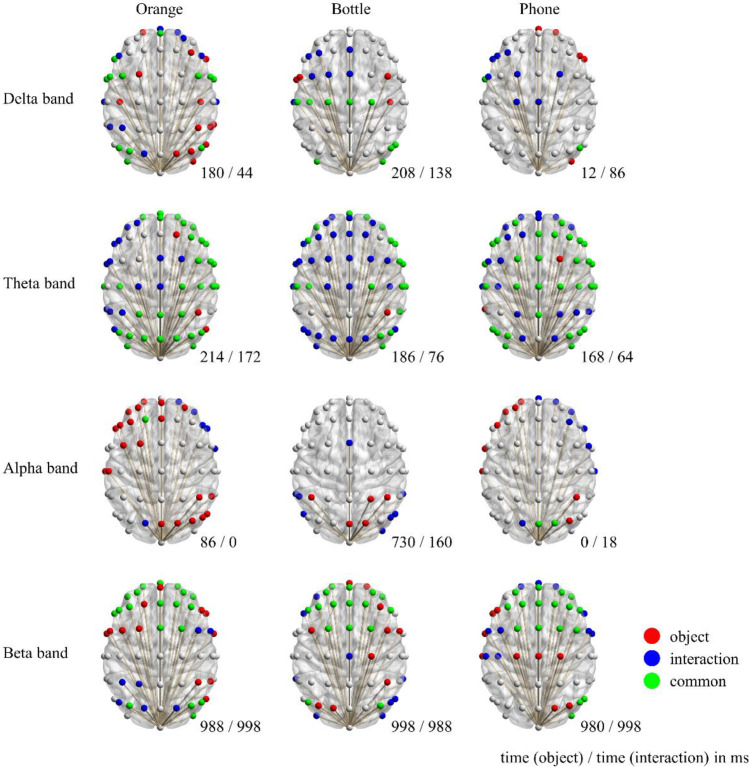
Functional connectivity between visual cortex and other regions. Colored electrode indicates that connectivity between that region and the occipital lobe actually exists. Time is indicated at the bottom right corner of each topography, as “time (ms) that most connectivity occurred when seeing objects/time (ms) that most connectivity occurred when seeing objects being grasped”. Note that each topography is an overlay of two graphs at two different moments. Red and blue electrodes represent the connections that only occurred when seeing objects and when seeing objects being grasped, respectively, while the green ones mean the two conditions share the same electrode.

In summary, the topography demonstrated that functional connectivity between the occipital lobe and regions of RF, L/RCS, RAG, and CF were established similarly when participants either saw images of objects or those of interactions. To make it more intuitive, the PLV-over-time plot of the regions mentioned above is shown in Figure 3. On the contrary, the difference is embodied in the area covered by electrodes F5, F7, FC5, and FT7, which is believed to be Broca’s area (BA) [28,29,30,31] and the LAG. Same as before, we demonstrated these differences in the plot of PLV over time in Figure 4. The results of paired test suggested that these differences are significant.

### 3.2. Power Variations

As mentioned in the Introduction, we were expecting to find some motor-related EEG features when participants looked at non-tool objects. Thus, our attention was turned to power changes in the mu rhythm [32,33,34], and clear event-related desynchronization (ERD) was noticed with both “seeing objects” and “seeing interactions”, as shown in Figure 5a. The topography was drawn with EEG data filtered at 8–13 Hz and then was Laplacian spatial filtered to highlight the changes. ERD was mainly observed at the region of the bilateral postcentral gyrus, which may suggest the participation of the primary somatosensory cortex [35]. Among all three objects, the most obvious ERD occurred at the area covered by electrodes C5, CP3, and CP5 in the left hemisphere (LS), as well as the corresponding position in the right hemisphere (RS). Figure 5b revealed its dynamic changes over time. Although all of these plots performed clear ERD at the end, there was obvious event-related synchronization (ERS) observed during the process when participants saw objects being grasped. This ERS was widespread from 100 to 200 ms, especially in LS.

## 4. Discussion

The purpose of this study was to investigate whether seeing manipulable objects would lead to a similar phenomenon to that when seeing tools. A previous research studied the difference between tools and “objects without manipulability” and reported that the stage of confirming whether a presented object is able to be operated happens in the first 250 ms after visual stimulus onset, and the conclusion leads to the activation of the left somatosensory cortex and the bilateral premotor cortex [36]. Moreover, they also mentioned that Brodmann areas 19 and 37 were activated in the ventral side, whether the object was manipulable or not. In our study, we noticed the functional connectivity peaked at 200 ms approximately between the visual cortex and RAG (BA39, border on BA19 and BA37) and between RF (close to the premotor cortex in the right hemisphere) and LCS (the left somatosensory cortex). However, we did not find enough evidence to imply the participation of the left premotor cortex. Additionally, our results indicated that RCS also joined the cognition process after seeing a manipulable object. Another study that paid attention to the mu rhythm ERD phenomenon when seeing tools found that it can be noticed as early as in the first 175 ms [37]. These spatial and temporal commonalities suggested the perception of a manipulable object is similar to those of tools.

Many studies considered that the particularity of tools is derived from the action applied to use them, which they come naturally with [38]. Therefore, we suspected that the presentation of a manipulable object may cause a similar cerebral activity to that which occurs upon seeing an interaction with that object. However, our experimental results rejected this inference with the additional functional connectivity between the occipital lobe and BA as well as between the occipital lobe and LAG when participants were shown images of objects being grasped. Although the controversy about its location is still on-going, a large majority of scholars believe that the mirror neuro system (MNS) exists near Broca’s area (or BA44), the inferior parietal lobule (near the LAG), and the superior temporal sulcus [39,40,41,42]. Hence, connectivity observed at BA and LAG can be reasonably regarded as activity of the MNS evoked by seeing the action of grasping objects. This may explain the different distributions of functional connectivity for seeing objects vs. seeing interactions with objects; nevertheless, ERS in the somatosensory cortex, which can only be noticed in the latter case, still exists. All of this evidence led us to the conclusion that the changes observed in the cerebral cortex after seeing objects being grasped were not the same as those that occurred after seeing only objects.

Undoubtedly, the difference in power change in the somatosensory cortex is due to the difference in visual stimuli, which means that the ERS may be caused by the hand contained within the image or the combination of a hand and the object. Fortunately, we have collected EEG data from when participants were shown only a hand and both a hand and an object. By comparing the topography in Figure 6a, we found that they showed ERS in the left somatosensory cortex for both conditions, although the values were not completely the same. This suggests that the hand seen in the visual stimuli partly contributed to the ERS. We also analyzed the data from condition C and interestingly found that it was different from that of the other three kinds of stimulus. It seems that participants recognized the hand and object in each image as two entities. We found that, at about 200 ms after visual stimulus onset, a positive event-related potential (ERP) component appeared at both the PO7 and PO8 electrodes but with a right hemisphere asymmetry. The plot in Figure 6b shows the ERP difference between the PO7 and PO8 electrodes. Evidently, two clear peaks were observed in condition C, while only one was observed in the other two conditions. A further test with a one-way ANOVA-based multiple comparison suggested that the laterization phenomenon in condition C was significantly different from the others (*p* < 0.05).

In summary, this study investigated the functional connectivity between the visual cortex and the other regions after healthy participants saw daily objects that are manipulable; we compared our results with those of previous studies regarding brain activity after seeing tools. We found that seeing manipulable objects and seeing tools caused similar phenomena in both time and space. Next, we assessed whether seeing a manipulable object led to a similar mu rhythm change to seeing an interaction with the same object; however, the evidence rejected our hypothesis: additional activation of Broca’s area and the left angular gyrus, and early alpha band ERS in the somatosensory cortex were only observed when participants saw interactions.

## Figures and Tables

**Figure 1 sensors-22-07771-f001:**
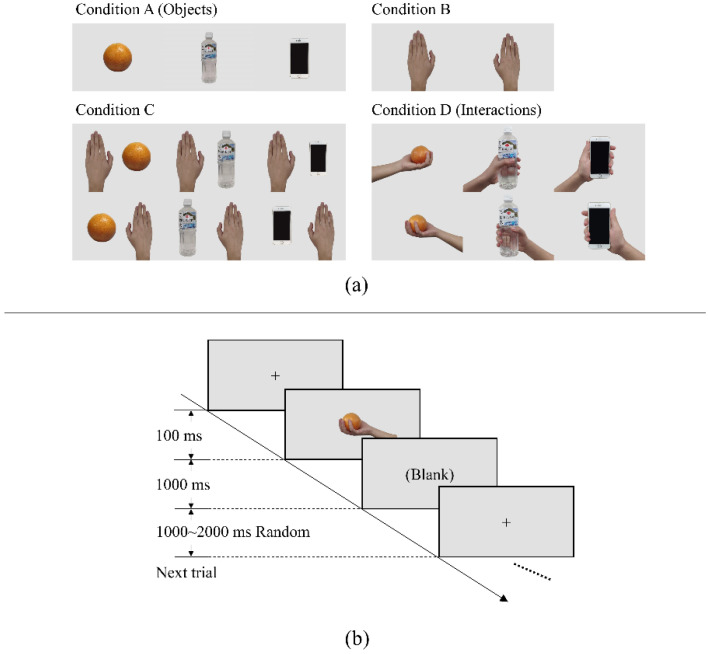
(**a**) Four kinds of images used in our experiment. Condition A presented participants with images of an orange, bottle, and smart phone (three objects). Condition B presented images of hands. Condition C combined the three objects and hands within the images. Condition D showed whole actions of hands grabbing objects (interactions). (**b**) Workflow of the trial. The images after the cross were randomly chosen from images corresponding to the current session (e.g., orange session, bottle session, and phone session).

**Figure 3 sensors-22-07771-f003:**
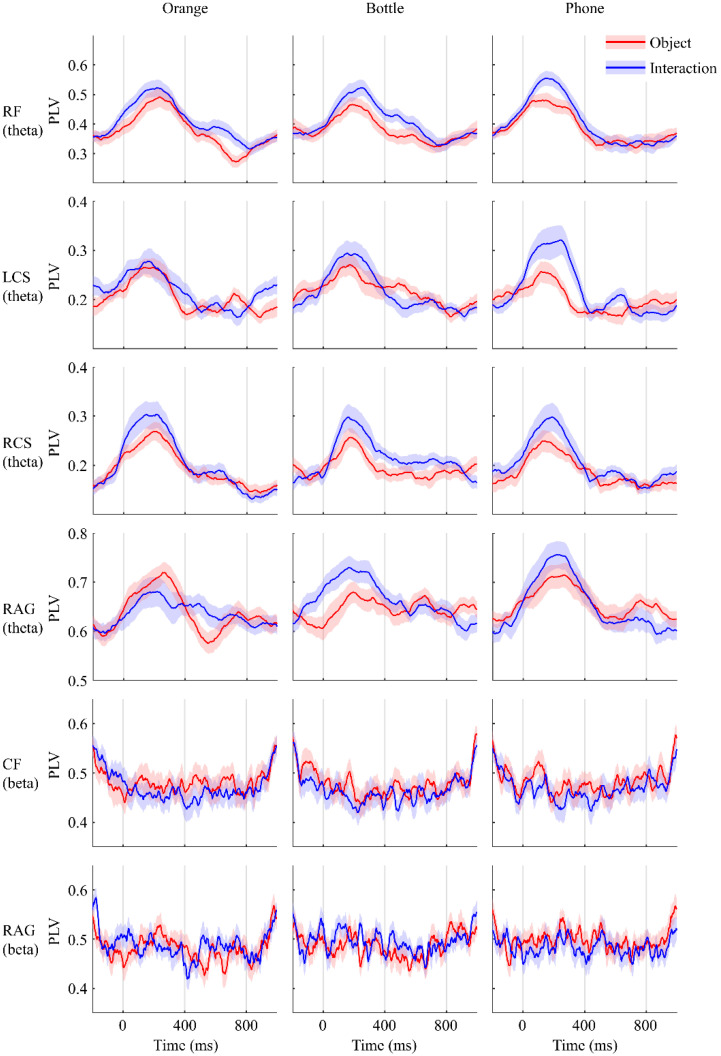
PLVs over time. Red line shows phase locking values (PLVs) when participants were shown objects, while the blue line shows PLVs when they were shown objects being grasped by human hands. Shaded areas are standard error. On these shown regions, PLVs from the two conditions varied similarly for the theta and beta bands.

**Figure 4 sensors-22-07771-f004:**
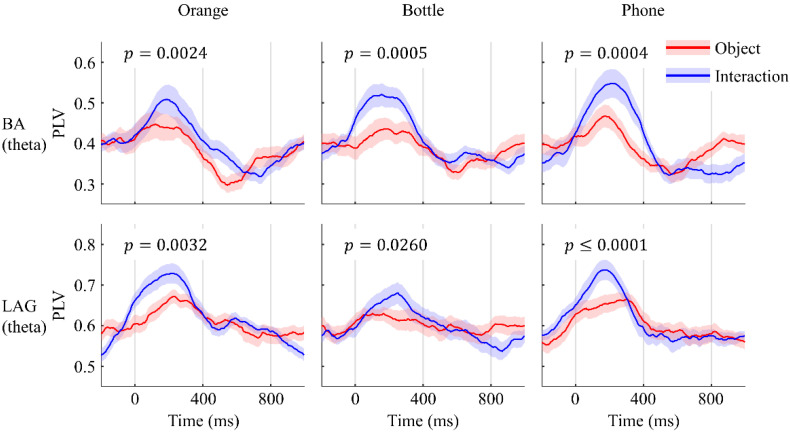
PLV observed at BA and LAG. Red line shows PLVs when participants were shown objects, while the blue line shows PLVs when they were shown objects being grasped by human hands. Shaded areas are standard error. Significant difference was noticed between seeing objects and seeing interactions at 200 ms after presenting the stimulus to participants (α = 0.05).

**Figure 5 sensors-22-07771-f005:**
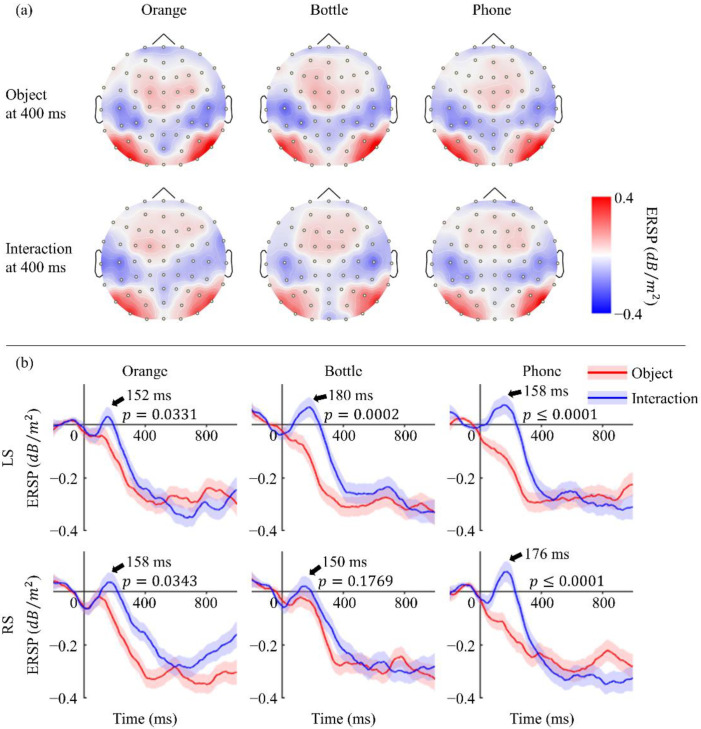
(**a**) Topography of ERSP at 400 ms. Mu rhythm ERD distributed at bilateral posterior central gyrus with a little left advantage and performed similarly in all six situations. (**b**) ERSP over time. Red line shows ERSP when participants were shown objects, while the blue line shows ERSP when they were shown objects being grasped by human hands. Shaded areas are standard error. A clear ERS was observed only when seeing interactions, and its peak time is indicated with an arrow. The significance of ERS was confirmed by a permutation test on the ERSP value in the two conditions at the corresponding time (α = 0.05).

**Figure 6 sensors-22-07771-f006:**
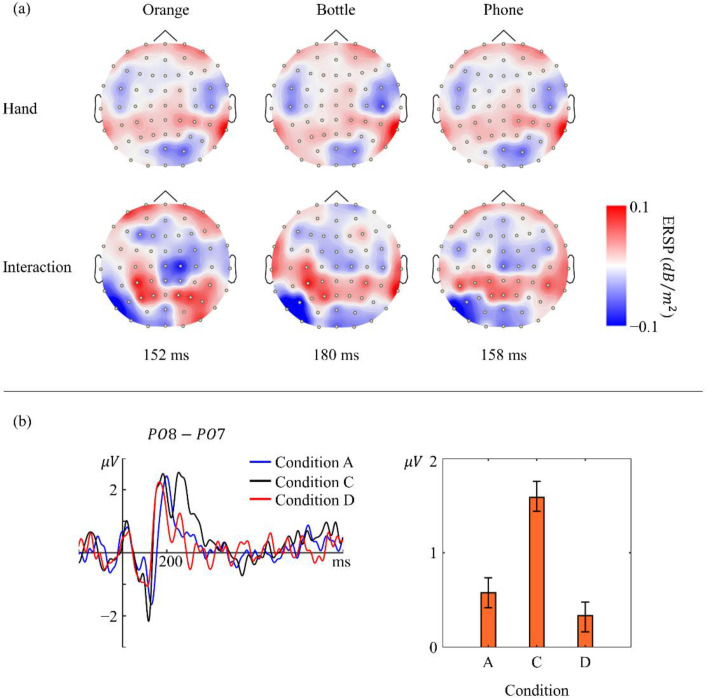
(**a**) Topography of 8–13 Hz ERSP when seeing human right hand and seeing interactions using the right hand at 152, 180, and 158 ms. ERS at LS is weaker when only images of a hand are presented to participants. (**b**) Plot shows a grand averaged ERP difference between electrodes PO7 and PO8. A remarkable second peak (black line) appeared when participants were presented with images in condition C. The bar graph on the right shows mean and standard error of the difference data in the range from 246 to 300 ms.

## Data Availability

The data presented in this study are available from the corresponding author upon request. The data are not publicly available due to restrictions for participant privacy.

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
