# Peer review of "Spatio-Temporal Neural Dynamics of Observing Non-Tool Manipulable Objects and Interactions"

_sensors, 2022, doi:10.3390/s22207771_

Round 1
Reviewer 1 Report
- Summary: This manuscript presents an interesting investigation of a human cognitive study based on EEG recording. The authors were trying to interrogate whether observing 1) a manipulable object or 2) a manipulable object grasped by a human hand elicits similar cerebral activity in the brain as the participant seeing images of tools. Here are my comments regarding the manuscript:
- Title: the whole sentence should be rephrased. The current title is too long and very hard to understand.
- Writing issue:
1) Typos: (line 36) "as least" -> "at least"; (line 43, 49 & 54) "seeing an object along" -> "seeing an object alone"?
2) Grammar errors: missing "the" all the time, as well as many grammar issues. I recommend that the authors use Grammar checking/correction software to inspect the text.
3) Suggestions: try to break down long sentences into short ones and connect them by tight logic.
- The experimental design: the authors claimed to compare functional connectivity between "seeing tools" and "seeing manipulable objects grasped by a human hand). Then why not include a condition showing images of tools in the current study? This way, the comparison can be more straightforward.
- The conclusions: Maybe I misunderstood some of the text, but I think the conclusions were not clearly presented. In Discussion, the authors stated: (line 258-259) "The purpose of this study was to investigate whether seeing objects grasped by a human hand would lead to a similar phenomenon as when seeing tools." Nevertheless, in their summary, the authors concluded: (line 316-321)
1) "We found that seeing manipulable objects and seeing tools caused similar phenomenon in both time and space." --- indicating that your Condition A (Objects) is similar to "seeing tools."
2) "we assessed whether seeing an manipulable object led to similar mu rhythm change as seeing as interaction with the same object, however the evidence rejected our hypothesis." --- indicating that your Condition A (Objects) is different from Condition D (interactions).
Where is the conclusion for comparison between Condition D and "seeing tools"?
Discussion session needs more work to clarify the results and conclusion.
Reviewer 2 Report
I enjoyed reading the paper. The authors have presented an interesting article about a Dynamic Functional Connectivity and Time Frequency Features between Visual Cortex and Other Brain Regions while Observing Manipulable Objects and Observing Interactions.
Abstract, overview
The abstract is a concise description of the work. The introduction is well structured, and it covers all the concepts investigated in the methodological part. The previous work is well presented and integrated. I consider that this work brings added value in the field and the specific objectives of the manuscript are well related to the previous work developed in this domain.
Methodology
The research design used is appropriate in order to answer the research questions proposed by the authors. The methods are described properly. The results are clearly presented and are in relation to the concepts investigated.
Discussion and conclusions
The discussions are clear and concise. The conclusions are strongly related to the findings of the research work.
Format and style
All the format and style features were respected and are compliant with the requirements.
References
The format of the reference list fixes well to the specified format.
Plagiarism and any other ethical concerns about this study
I do not have any potential conflict of interest with regards to this paper.
Despite the good work done, there is still some room for improvement, as follows:
- I think some more literatures should be added. Besides the mentioned systems there are several others (like cost-effect BCI, eye-tracking, VR/AR) which are applied nowadays. It would be good to see the "effect of different web-based media" content on "human brain waves", as well as the additional applications of brainwave-based control like in examine the effect of different web-based media on human brain waves. It would improve the quality of the publication to mention the relationship between a cognitive psychological attention test and the attention levels determined by a BCI systems such as in an examination and comparison of the EEG based attention test with CPT and TOVA. In addition to BCI systems, mentioning other important human-computer interaction eye movement tracking would also improve quality, as such systems can be used in the analysis of programming technologies such as LINQ and algorithms, thus enabling, for example, cognition load or source code, algorithm description tools readability testing like in measuring cognition load using eye-tracking parameters based on algorithm description tools, in clean and dirty code comprehension by eye-tracking based evaluation using GP3 eye tracker and in analyse the readability of LINQ Code using an eye-tracking-based evaluation or VR in hand controlled mobile robot applied in virtual environment, or in a review of human–computer interaction and virtual reality research fields in cognitive InfoCommunications.
Author Response
Thank you for your distinctive comments. It is sensible to connect the current study with applicable systems, like BCI or eye-tracking operation system. Actually, we have already been working on this topic. In another still on-going project, we are trying to classify human’s brain activity when seeing other’s movement, recalling a movement, and actually executing a movement, via EEG. We think this research would be helpful in the scenario of human-machine coordination. What we found in the current study provided a possible solution to finish the new on-going one, and moreover, the methods mentioned in this manuscript will be used in that research as well, for extracting spatio-temporal features that can be useful in the pattern recognition part of the classification job. It is my plesure to have similar future vision with you, and we will take into account your comments in the new research.
Round 2
Reviewer 1 Report
Thanks for the prompt responses.
I'm still not satisfied with the title: Spatio-temporal EEG Features while Observing Non-tools Manipulable Objects and Interactions
How about: Spatio-temporal neural dynamics of observing non-tools manipulable objects and interactions
Besides the title, I have no further questions.
Author Response
Thanks for your suggestion.
I think the title you proposed is more appropriate indeed, so I revised the title into "Spatio-temporal Neural Dynamics of Observing Non-tools Manipulable Objects and Interactions".
Reviewer 2 Report
Despite the good work done, there is still some room for improvement, as follows:
- How was the data validated?
- What improvement does the developed system show with those already published?
- I think some more literatures should be added. Besides the mentioned systems there are several others (like cost-effect BCI, eye-tracking, VR/AR) which are applied nowadays. It would be good to see the "effect of different web-based media" content on "human brain waves", as well as the additional applications of brainwave-based control like in examine the effect of different web-based media on human brain waves. It would improve the quality of the publication to mention the relationship between a cognitive psychological attention test and the attention levels determined by a BCI systems such as in an examination and comparison of the EEG based attention test with CPT and TOVA. In addition to BCI systems, mentioning other important human-computer interaction eye movement tracking would also improve quality, as such systems can be used in the analysis of programming technologies such as LINQ and algorithms, thus enabling, for example, cognition load or source code, algorithm description tools readability testing like in measuring cognition load using eye-tracking parameters based on algorithm description tools, in clean and dirty code comprehension by eye-tracking based evaluation using GP3 eye tracker and in analyse the readability of LINQ Code using an eye-tracking-based evaluation or VR in hand controlled mobile robot applied in virtual environment, or in a review of human–computer interaction and virtual reality research fields in cognitive InfoCommunications
Round 3
Reviewer 2 Report
Accept in present form.